# Illumination Distribution Prior for Low-light Image Enhancement

## ABSTRACT

In this paper, we propose a simple but effective illumination distribution prior (IDP) for images to illuminate the darkness. The illumination distribution prior is the product of a statistical approach to low-light images. It is based on a key factor - the mean value and standard deviation of images are positively correlated with the illumination. Using IDP in combination with the dual-domain feature fusion network (DFFN), we can obtain images that are more consistent with the ground truth distribution. DFFN inserts the discrete wavelet transform (DWT) into the transformer architecture, aiming to recover the detailed texture of the image through local high-frequency information and global spatial information. We have conducted extensive experiments on five widely used low-light image enhancement datasets and the experimental results show the superior performance of our proposed network (IDP-Net) compared to other state-of-the-art methods.

## CCS CONCEPTS

• **Computing methodologies** → **Computer vision problems**.

## KEYWORDS

Low-light image enhancement, Illumination distribution prior, Local high-frequency information, Global spatial information

## 1 INTRODUCTION

Various types of imaging devices provide a convenient means to capture exquisite images of our daily lives. However, under the constraints of low-light conditions, the acquired images usually have undesirable conditions such as low contrast, color distortion and noise amplification, which greatly affects the visual experience of the images and challenges advanced computer vision tasks such as face recognition [63] and object detection [37] at night. Thus, low-light image enhancement aims to recover the information hidden in low-light images, such as illumination and texture, and thus improve the quality of the images.

In light of this, numerous researchers have proposed various methods for low-light image enhancement. For plain methods, histogram equalization [42] and gamma correction [19] enhance the contrast of the image but often produce unwanted artefacts. Also, traditional methods are based on Retinex theory [2, 25], which decomposes the image into two components, reflectance and illumination, and recovers them separately. However, the enhanced image appears with severe noise and local color distortion.

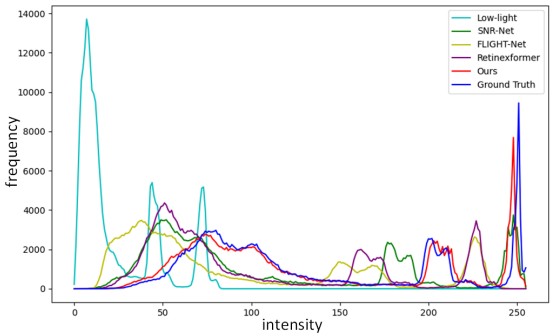

(a) Illumination Distribution

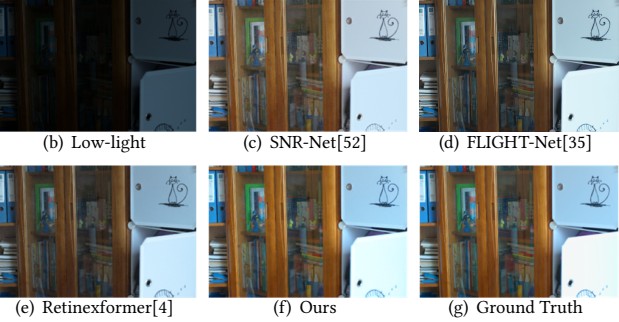

(b) Low-light    (c) SNR-Net[52]    (d) FLIGHT-Net[35]

(e) Retinexformer[4]    (f) Ours    (g) Ground Truth

**Figure 1: Compared to state-of-the-art architectures, our method is closer to the ground truth distribution, where (a) represents the distribution of the different methods and (b)-(g) represent the corresponding visualization results.**

With the rapid development of deep learning, convolutional neural networks are widely used in the field of low-light image enhancement and have made significant progress. These CNN-based algorithms are usually divided into two categories: deep retinex-based decomposition and end-to-end mapping. Deep retinex decomposition [49, 54] enhances low-light images through illumination map estimation or reflectance recovery. End-to-end mapping [12] learns the mapping of low-light images to ground truth through codecs or convolutional blocks. However, these methods tend to ignore the issue of image illumination distribution, which leads to the probability of local overexposure during the recovery process, and even undesired conditions such as color distortion and noise amplification.

In addition, some existing methods cannot extract high frequency information well. Fan *et al.* [10] used discrete wavelet transform to obtain the output features of wavelet attention feature information. However, they did not distinguish between high-frequency and low-frequency information when extracting wavelet domain features, resulting in not highlighting the features of high-frequency information. Besides, discrete wavelet transform focuses more on

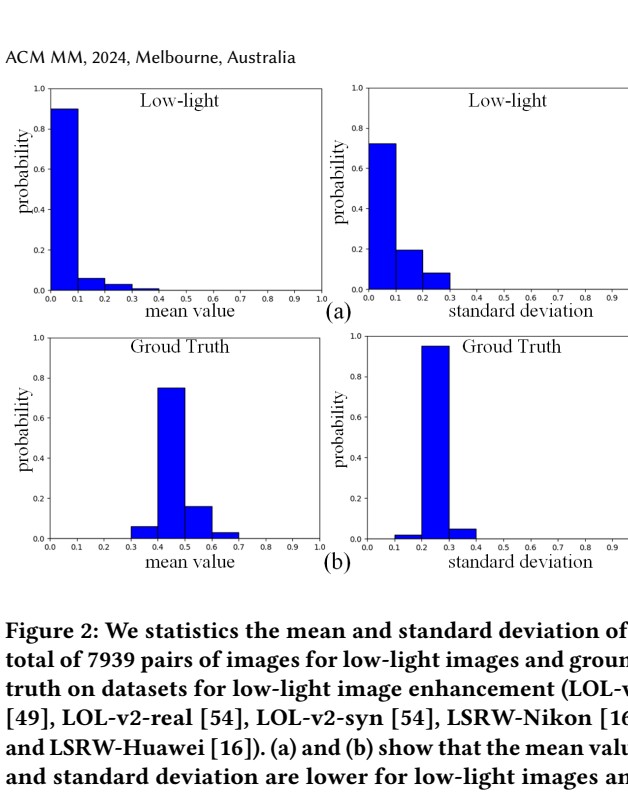

Figure 2: We statistics the mean and standard deviation of a total of 7939 pairs of images for low-light images and ground truth on datasets for low-light image enhancement (LOL-v1 [49], LOL-v2-real [54], LOL-v2-syn [54], LSRW-Nikon [16], and LSRW-Huawei [16]). (a) and (b) show that the mean value and standard deviation are lower for low-light images and higher for ground truth.

the characterization of local information and cannot effectively capture global features.

**Our solution for low-light image enhancement.** Inspired by related work of layer normalization [1, 23, 41, 55], the magnitude of the mean and standard deviation of an image is correlated with the illumination. Based on this, we propose illumination distribution prior to remove darkness from low-light images. The illumination distribution prior is based on the statistics of the mean value and standard deviation of the images. As shown in Fig. 2 (a) and (b), we observe that the mean value and standard deviation of the low-light images are lower, while those of the ground truth are higher. Thus, the illumination distributions of low-light images and ground truth are different. Notably, Fig. 3 (a) shows that the illumination distributions are not the same for different sub-regions (same scale) of the same image. Fig. 3 (b) shows that the illumination distributions of sub-regions (similar backgrounds) with different scale sizes are also different. Considering the above, the illumination distribution prior restores the illumination by migrating the illumination distributions at multiple scales, which is beneficial for generating images that are closer to the ground truth distribution. Also, using discrete wavelet transform in combination with the transformer architecture, our proposed dual-domain feature fusion network (DFFN) is able to capture diverse dual-domain features to recover the detail texture. Specifically, DFFN is a U-shaped [40] network consisting of $conv3 \times 3$ and dual-domain feature fusion module (DFFM). DFFM ingeniously integrates the discrete wavelet transform into the transformer architecture, enhancing the recovery of intricate textures. This integration facilitates the collaborative extraction of information from both the local frequency domain and the global spatial domain, thereby optimizing the preservation and reconstruction of image details.

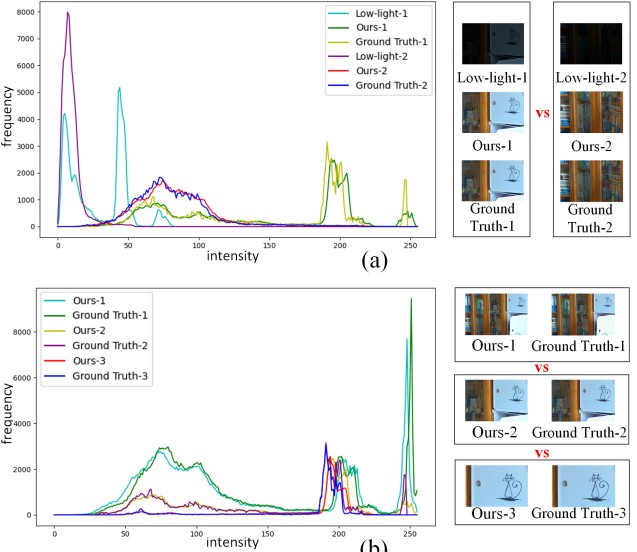

Figure 3: (a) selects two groups of images with the same sub-region size but with different backgrounds to show that different sub-regions have different illumination distributions. Note that "Low-light-x", "Ours-x", and "Ground Truth-x" (x = 1, 2) represent low-light images, our method, and ground truth, respectively. (b) selects three groups of images with different sub-region sizes but with similar backgrounds to show that there are differences in the illumination distributions of sub-regions at different scales. Note that "Ours-y", "Ground Truth-y" (y=1, 2, 3) represent our method and ground truth, respectively. Significantly, (a) and (b) also indicate that our method is closer to the distribution of ground truth.

Extensive experiments were conducted on five widely used low-light image enhancement datasets. The experimental results show that the enhanced images of our method are closer to the distribution form of ground truth and can adapt to different datasets and environments. In addition, the performance of our method consistently outperforms other state-of-the-art methods.

Overall, our contributions can be summarized as follows:

- We propose a novel illumination distribution prior for low-light image enhancement by constructing the multi-scale illumination distribution migration.
- We design a dual-domain feature fusion network, which integrates the local frequency-domain information of the discrete wavelet transform and the global spatial information of the transformer architecture, and utilises the rich dual-domain information for fine-grained restoration of texture structure and noise removal.
- Compared to other state-of-the-art competitors, quantitative and qualitative experiments validate that our architecture outperforms the SOTA method on five datasets.

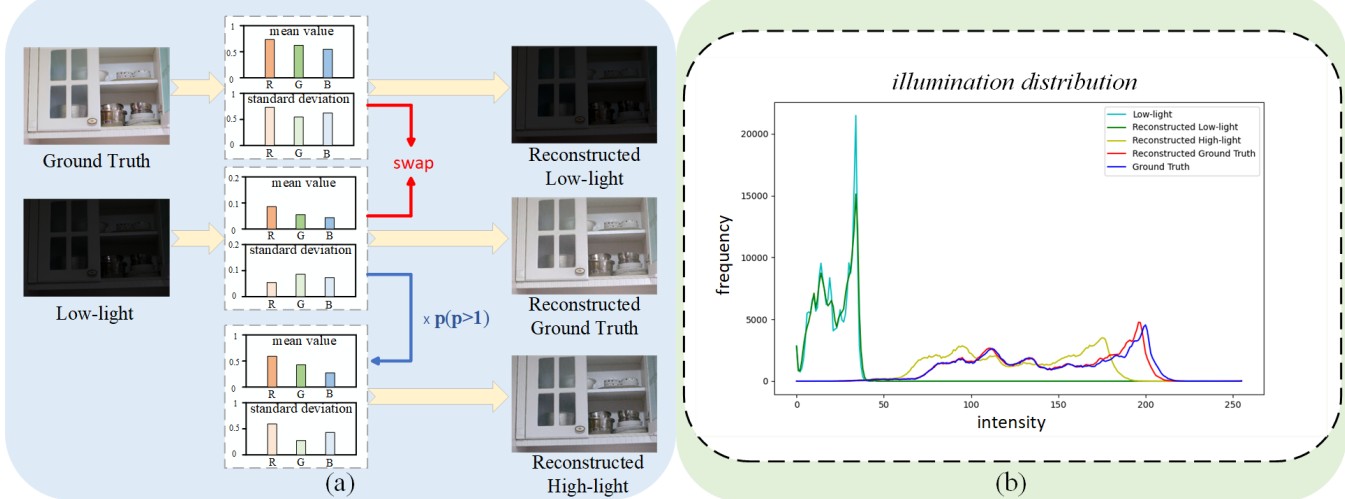

**Figure 4: Our motivation about IDP. In (a), we swap the mean value and standard deviation of each channel of a set of images with the same background but different illumination conditions ("Low-light" and "Ground Truth") to obtain the swapped results "Reconstructed Low-light" and "Reconstructed Ground Truth". The mean value and standard deviation of the low-light image are scaled by a constant value $p$ , which can generate "Reconstructed High-light". From visual perspective, the mean value and standard deviation represent the magnitude of the image brightness. In (b), we use grayscale histogram to represent the illumination distributions of "Low-light", "Reconstructed Low-light", "Ground Truth", "Reconstructed Ground Truth", and "Reconstructed High-light". The distributions of "Reconstructed Ground Truth" and "Reconstructed High-light" are close to "Ground Truth", while the distribution of "Reconstructed Low-light" is close to "Low-light".**

## 2 RELATED WORK

### 2.1 Low-Light Image Enhancement

Histogram equalization [20, 27] and gamma correction [5] are used to improve image contrast by enlarging the dynamic range, which ignored the structural information of the images and usually produced unwanted artifacts. Traditional methods [29] are based on the retinex theory to recover reflectance and illumination separately to obtain enhanced results. Guo *et al*. [15] proposed to improve the initial illumination maps of low-light images through structural prior to achieve low-light enhancement. However, these methods lead to color distortion and noise.

With the development of deep learning, many learning-based [3, 13, 16, 21, 24, 30, 36, 48, 60, 62] low-light image enhancement methods have been proposed in recent years. Guo *et al*. [13] designed a new pixel-level and higher-order curve that can be adjusted in dynamic range for unsupervised training. Zhang *et al*. [59] generated more natural and colorful post-enhancement images by learning the structural textural information and the color distribution information of the images, respectively. Ozcan *et al*. [35] learned the individual gain coefficients of the illumination maps from low-light images and multiplied them with the low-light maps to recover the images. However, these methods lack local perception of regions and remote dependencies, making it difficult to capture illumination distributions and texture details.

### 2.2 Prior Knowledge

Prior knowledge [6, 14, 39, 43, 50] can provide unique information from degraded images to assist in image restoration. He *et al*. [17] found that most local patches contain some pixels with very low intensity in at least one color channel, and proposed the dark channel prior for image haze removal. Jin *et al*. [22] proposed the deep inconsistency prior to guide RGB-NIR fusion with the help of structural inconsistency. Based on the powerful feature representation capability of Masked Autoencoder (MAE), zheng *et al*. [60] proposed MAE-based illumination and noise prior for low-light image enhancement. Therefore, reasonable use of prior knowledge is necessary.

### 2.3 Vision Transformer

Vaswani *et al*. [45] first proposed Transformer for processing machine translation tasks. Later, transformer was widely used in computer vision tasks *et al*. [4, 9, 32, 33, 41, 52] and achieved remarkable results. Xu *et al*. [52] proposed a signal-to-noise ratio-aware transformer and convolutional model for low-light image enhancement. Cai *et al*. [4] utilized the illumination information to guide the transformer to establish remote dependencies. However, the transformer architecture acquires the single spatial domain information and does not recover well for some complex textures.

### 2.4 Frequency Domain Information

In recent years, frequency domain information has received attention from many researchers and has been effectively verified in several computer vision fields [10, 11, 18, 31, 44, 51, 57, 61]. For example, Tian *et al*. [44] constructed a multi-stage image denoising model using wavelet transform. Liu *et al*. [31] used a wavelet-based two-branch network for image de-rain. Fu *et al*. [11] designed a

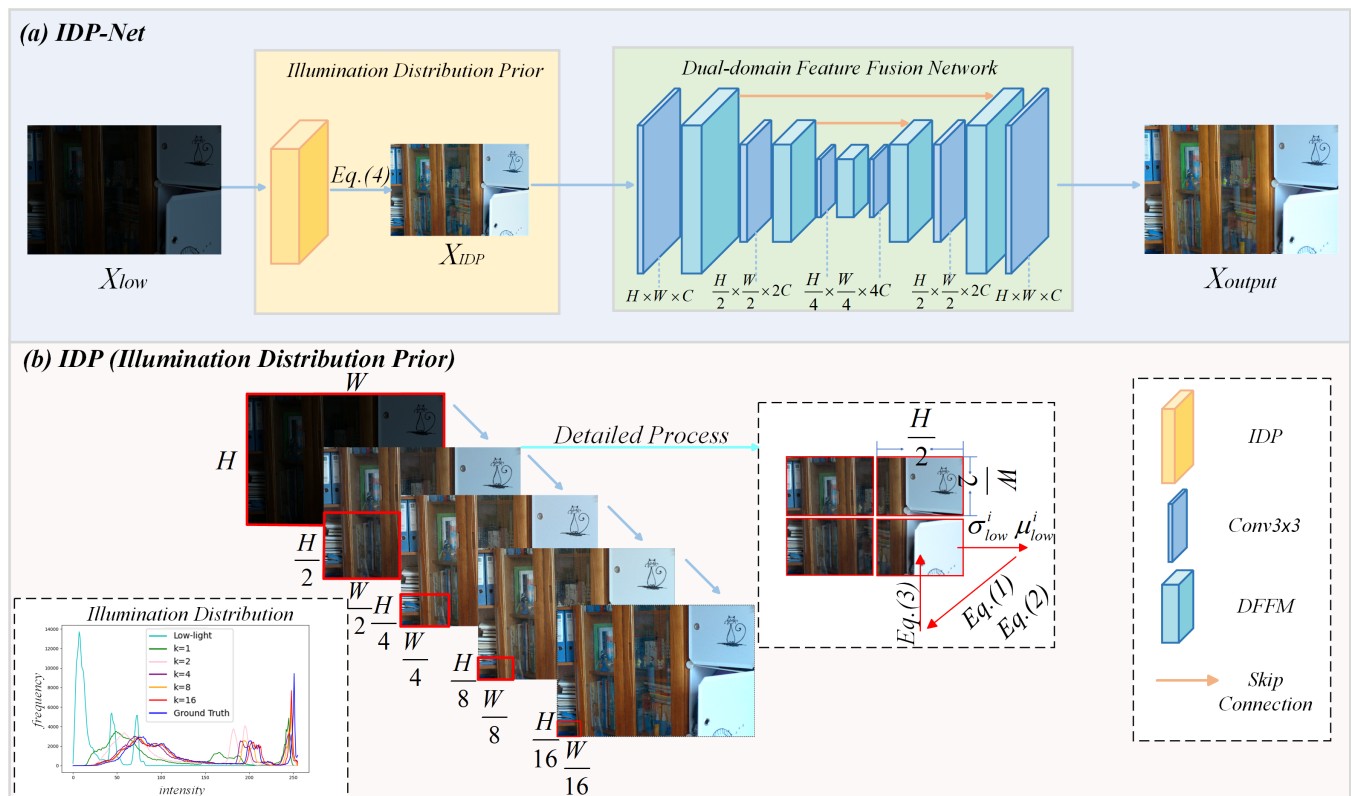

**Figure 5: (a) shows the overall architecture of our method. (b) shows the complete process of IDP. Also, the evolutionary illuminance distribution process of IDP is illustrated in (b). In particular, we enumerate the detailed process of IDP when k is 2.**

DWT branch for image defogging to extract high frequency information.

The above methods show that the discrete wavelet transform can effectively extract high-frequency information from images and significantly improve the performance of image recovery. Therefore, we insert the discrete wavelet transform into the transformer architecture to obtain rich information in both frequency and spatial domains. In this way, our model can utilize the dual-domain information to effectively recover the detailed texture of the image.

## 3 METHOD

### 3.1 Motivation for Illumination Distribution Prior

Our motivation is shown in Fig. 4. First, given a set of paired "Low-light" and "Ground Truth" images, we derive the corresponding mean value and standard deviation. Subsequently, we swap the mean value and standard deviation for each channel of the "Low-light" and "Ground Truth". The swapped results show that the lighting conditions of both are transformed and the distribution of the "Reconstructed Ground Truth" is close to "Ground Truth". Then, appropriately increasing the magnitude of the mean and standard deviation of the low-light image, we obtain the brightened image "Reconstructed High-light" which is close to the illumination distribution of the "Ground Truth". Hence, we conclude that by appropriately increasing the mean value and standard deviation

of the low-light image, not only the illumination can be improved, but also the distribution of the ground truth is satisfied.

### 3.2 Darkness Removal Using Illumination Distribution Prior

Fig. 4 shows that appropriately increasing the mean value and standard deviation improves the low-light image illumination and satisfies the ground truth distribution. In addition, we are concerned about the differences (in Fig. 3 ) in illumination distributions (same scale, but different sub-regions or different scales, but similar sub-regions). Therefore, illumination distribution migration is achieved by increasing the mean value and standard deviation of the low-light image on sub-regions at different scales. By constructing the multi-scale illumination distribution migration, the distribution of the enhanced image on the sub-regions of different scales can be made closer to the distribution of ground truth, further reducing the error generated during the illumination migration process. We refer to the process of the multi-scale illumination distribution migration as illumination distribution prior (IDP).

To achieve the goal of illumination distribution prior removing the darkness, we repeatedly divide the input $X_{low} \in \mathbb{R}^{H \times W \times C}$ into $k^2$ sub-regions of the same size, where the sub-region size is $\frac{H}{k} \times \frac{W}{k}$ ($k = 1, 2, 4, 8, 16$). Then, on each sub-region, we perform the migration of the illumination distribution separately. Noticeably, we divide $X_{low}$ five times.

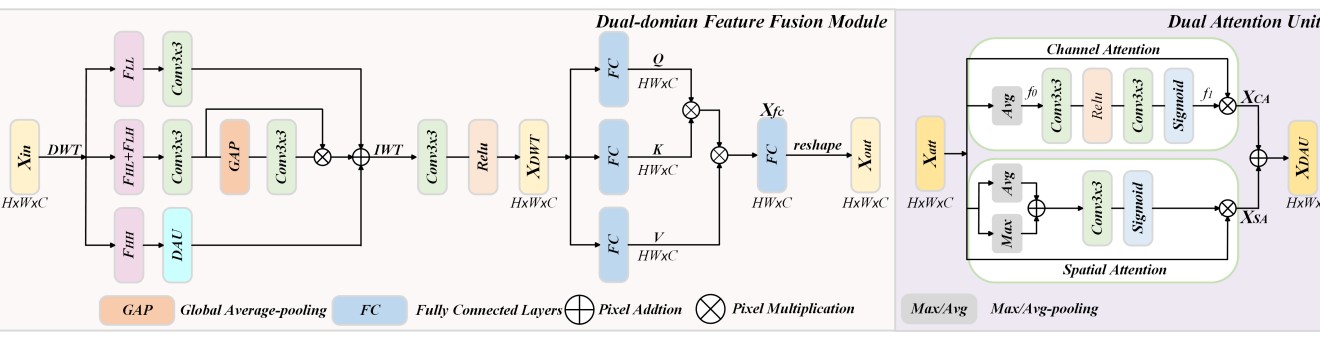

**Figure 6: Dual-domain Feature Fusion Module on the left and Dual Attention Unit (DAU) on the right. Notably, DWT and IWT represent the discrete wavelet transform and inverse wavelet transform.**

Specifically, we derive the mean value $\mu_{low}^i \in \mathbb{R}^{C \times 1}$ and standard deviation $\sigma_{low}^i \in \mathbb{R}^{C \times 1}$ for each channel of each sub-region. Then, to estimate the transform mapping of the mean value and standard deviation, we set two learnable parameters $W_\mu^i$, $W_\sigma^i \in \mathbb{R}^1$ as the recovered weights of $\mu_{low}^i$ and $\sigma_{low}^i$, respectively, and two learnable parameters $B_\mu^i$, $B_\sigma^i \in \mathbb{R}^1$ as the biases, respectively. The process of transformation is expressed as:

$$\mu_{high}^i = W_\mu^i \mu_{low}^i + B_\mu^i, \tag{1}$$

$$\sigma_{high}^i = W_\sigma^i \sigma_{low}^i + B_\sigma^i, \tag{2}$$

where $i$ represents the channel ($i = R, G, B$). $\mu_{high}^i \in \mathbb{R}^{C \times 1}$ and $\sigma_{high}^i \in \mathbb{R}^{C \times 1}$ represent the mean value and standard deviation of the corresponding channel of the enhanced image, respectively.

Then $\mu_{low}^i$ is subtracted from channel $i$ of each sub-region to remove the effect of the original low-light illumination. The illumination trend of the enhanced image is then constructed by multiplying $\sigma_{high}^i$ and removing $\sigma_{low}^i$ and adding $\mu_{high}^i$ to increase the illumination. In this way, the migration of the illumination distribution is achieved and can be expressed as:

$$X^{\frac{H}{k} \times \frac{W}{k}} = \frac{\sigma_{high}^i}{\sigma_{low}^i} \left( X_C^i - \mu_{low}^i \right) + \mu_{high}^i, \tag{3}$$

Using a non-overlapping displacement operation, we recover the illumination distribution on each sub-region to obtain the illumination distribution of the whole image. The process of obtaining the illumination distribution of the whole image in a single pass is denoted as $f^{\frac{H}{k} \times \frac{W}{k}}$.

Therefore, the output feature $X_{IDP} \in \mathbb{R}^{H \times W \times C}$ of IDP can be expressed as:

$$X_{IDP} = f^{\frac{H}{16} \times \frac{W}{16}} \left( f^{\frac{H}{8} \times \frac{W}{8}} \left( f^{\frac{H}{4} \times \frac{W}{4}} \left( f^{\frac{H}{2} \times \frac{W}{2}} \left( f^{H \times W} \right) \right) \right) \right), \tag{4}$$

where $f^{\frac{H}{k} \times \frac{W}{k}} \in \mathbb{R}^{H \times W \times C}$ ($k = 1, 2, 4, 8, 16$). Overall, IDP removes the darkness and creates the prerequisites for texture recovery.

### 3.3 Dual-domain Feature Fusion Network

**Network Structure.** Dual-domain Feature Fusion Network adopts a U-shaped [40] structure, which consists of $conv3 \times 3$ and dual-domain feature fusion module (DFFM). Given an input feature

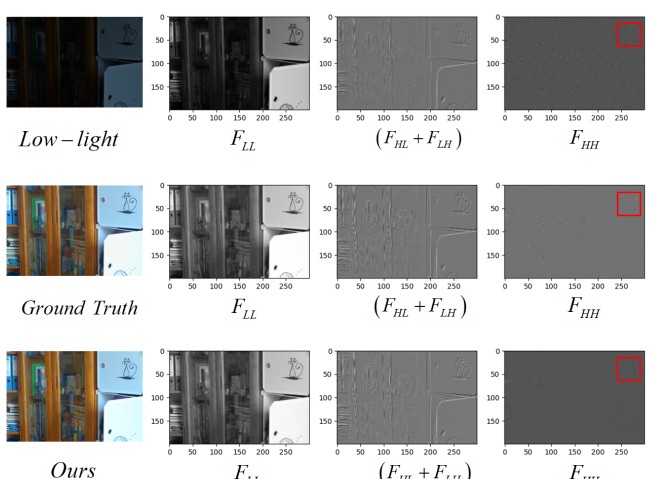

**Figure 7: Wavelet decomposition of low-light (the first row), ground truth (the second row), and ours (the third row) yields the frequency sub-bands $F_{LL}$, ($F_{HL} + F_{LH}$) and $F_{HH}$. The red boxes circle the features contained in the high-frequency sub-bands (zoomed-in viewing is recommended). The results of the comparison show that our network can acquire the high-frequency information well and recover the detail texture.**

$X_{IDP} \in \mathbb{R}^{H \times W \times C}$, $X_{IDP}$ is mapped into three different scales by two downsampling. Specifically, $X_{IDP}$ is extracted by $conv3 \times 3$ to obtain the feature map. Subsequently, the feature map undergoes sequential DFFM, $conv3 \times 3$ steps to implement the downsampling operation. where $conv3 \times 3$ is used to downscale the features. Then, the lowest resolution layer features obtained after downsampling are upsampled according to the symmetric structure of downsampling. In addition, we add some extra skip connections in the architecture for mitigating the loss of information. Finally, we obtain the enhanced image $X_{output} \in \mathbb{R}^{H \times W \times C}$.

**Discussion for DWT.** High-frequency information helps to improve the recovery of image details. DWT-FFC [61] applies the

discrete wavelet transform to extract wavelet features in the image defogging task, and the sub-bands generated by the discrete wavelet transform are considered to be of two categories: the low-frequency sub-band ($F_{LL}$) and the high-frequency sub-band ($F_{HL} + F_{LH} + F_{HH}$). However, this classification may not be suitable for low-light scenes. Fig. 7 shows the features of different frequency sub-bands after discrete wavelet transform decomposition. We observe that the features of the low-frequency sub-band of the low-light image ($F_{LL}$) have low illumination, and the structural features of the mixed high and low-frequency sub-band ($F_{HL} + F_{LH}$) are relatively clear. However, the features of the high-frequency sub-band ($F_{HH}$) of the low-light image are almost absent.

**Dual-domain Feature Fusion Module.** We focus on high frequency information by assigning different weights to sub-bands of different frequencies. Specifically, we employ the discrete wavelet transform (DWT) to decompose the input features $X_{in} \in \mathbb{R}^{H \times W \times C}$ into three categories, low-frequency sub-band ($F_{LL}$), mixed high and low-frequency sub-band ($F_{HL} + F_{LH}$), and high-frequency sub-band ($F_{HH}$). Next, the image is illuminated due to the assistance of the illumination distribution prior. Therefore, we use a $conv1 \times 1$ to extract the features of the low-frequency sub-band ($F_{LL}$) without additional illumination processing. As for the mixed high and low-frequency sub-band ($F_{HL} + F_{LH}$), we used a block consisting of $conv3 \times 3$ and global average pooling to extract the high-frequency information in it and moderate the relatively unimportant low-frequency information. In particular, we use a $DAU$ to emphasize the important features of the high-frequency sub-band ($F_{HH}$). $DAU$ divides the high-frequency sub-bands into two branches of spatial and channel dimensions to highlight key feature regions and channels, Spatial Attention ($SA$) and Channel Attention ($CA$). The $X_{att} \in \mathbb{R}^{H \times W \times C}$ is fed into the $DAU$ and the $SA$ applies parallel global average pooling and global maximum pooling to highlight regions of features along the channel dimension. The spatial attention map is then generated by a convolutional layer, followed by multiplication of the mapping normalized by the activation function by $X_{att}$ to obtain the output $X_{SA} \in \mathbb{R}^{H \times W \times C}$. $CA$ applies global average pooling to extrude $X_{att}$ into the feature $f_0 \in \mathbb{R}^{1 \times 1 \times C}$, which is passed through the convolutional layer and the activation function to obtain the weight value $f_1 \in \mathbb{R}^{1 \times 1 \times C}$. Subsequently $f_1$ is weighted and scaled by multiplication on $X_{att}$ to obtain the final output of the channel attention mapping $X_{CA} \in \mathbb{R}^{H \times W \times C}$. At the end of the $DAU$, the $CA$ and $SA$ sum the learned rich feature information and undergo a $conv1 \times 1$ to obtain the emphasized features of the high-frequency sub-band $X_{DAU} \in \mathbb{R}^{H \times W \times C}$. More, we integrate the weighted wavelet features of the three frequency sub-bands and apply the inverse wavelet transform (IWT). Finally, the wavelet transform optimized feature $X_{DWT} \in \mathbb{R}^{H \times W \times C}$ is obtained after $conv3 \times 3$.

Wavelet feature $X_{DWT} \in \mathbb{R}^{H \times W \times C}$ passed into the multi-head self-attention part. We chose to remove the illumination information in the multi-head self-attention part of Retinexformer [4], taking into account the positive effect of our illumination distribution prior on illumination. Firstly, the feature $X_{DWT}$ is reshaped into $X \in \mathbb{R}^{HW \times C}$, and then $X$ is divided into $m$ heads:

$$X = [X_1, X_2, \cdots, X_m], \tag{5}$$

where $X_n \in \mathbb{R}^{HW \times \frac{C}{m}}$ and $n = 1, 2, \cdots, m$. For each header, $X_n$ is linearly mapped to the query element $Q_n \in \mathbb{R}^{HW \times \frac{C}{m}}$, the key element $K_n \in \mathbb{R}^{HW \times \frac{C}{m}}$, and the value element $V_n \in \mathbb{R}^{HW \times \frac{C}{m}}$ as using a fully connected layer:

Thus, the self-attention of each head can be expressed as:

$$Attention\,(Q_n,\ K_n,\ V_n) = softmax\left(\frac{K_n^T Q_n}{\alpha_n}\right) V_n, \tag{6}$$

where $\alpha_n$ is a learnable parameter. Next, $m$ heads are concatenated to produce the output feature $X_{fc} \in \mathbb{R}^{HW \times C}$ by a fully connected layer. Finally, we reconstruct $X_{fc}$ to obtain the output feature $X_{out} \in \mathbb{R}^{H \times W \times C}$ of the multi-head self-attention.

## 4 EXPERIMENT

### 4.1 Datasets and Implementation Details

**Dataset.** We evaluate the proposed method on five widely used datasets for low-light image enhancement: LOL-v1 [49], LOL-v2-real [54], LOL-v2-syn [54], LSRW-Nikon [16], and LSRW-Huawei [16]. The LOL dataset consists of two versions: LOL-v1 and LOL-v2. LOL-v1 and LOL-v2-real are collected from real-world scenarios. LOL-v1 contains 485 pairs of low-light/ground truth images for training and 15 pairs for testing. LOL-v2-real contains 689 pairs for training and 100 pairs for testing. LOLv2-syn is generated by synthesizing low-light images from RAW images based on the analysis of illumination distributions. It includes 900 pairs for training and 100 pairs for testing. LSRW-Huawei and LSRW-Nikon are captured using Huawei P40 Pro and Nikon D7500 cameras, respectively, in real-world scenes. The training and testing sets of LSRW-Huawei and LSRW-Nikon are divided in a ratio of 3150:30 and 2450:20, respectively.

**Implementation Details.** We implement the our model in PyTorch [38] and train it on an NVIDIA 3090 GPU. The model is trained for $2.5 \times 10^5$ iterations using the Adam [26] optimizer with the momentum $\beta 1 = 0.9$ and $\beta 2 = 0.999$. Additionally, cosine annealing scheme [34] is employed to adjust the learning rate of model during training. The learning rate is initially set to $2.0 \times 10^{-4}$ and gradually decreased to $1.0 \times 10^{-6}$. To improve experimental performance, data augmentation is applied by randomly rotating and flipping the images during training. Furthermore, the training samples are cropped into patches of size $128 \times 128$, and the batch size is set to 8. For the loss function, we choose the $\mathcal{L}1$ loss function to constrain the minimum difference between the output images of our method and the ground truth, which can be expressed as:

$$L_{loss} = \frac{1}{N} \sum_{i=1}^{N} \left\| X_{gt} - X_{output} \right\|_1, \tag{7}$$

where $N$ represents the number of samples for training, and $\|\cdot\|_1$ denotes the $\mathcal{L}1$ norm.

### 4.2 Comparison with State-Of-The-Arts

In this paper, we compare our method with the state-of-the-art low-light image enhancement methods in recent years, including SID [7], KinD [58], LPNet [28], Band [53], IPT [8], Sparse [54], HDMNet [30], MIRNet [56], SNR-Net [52], FourLLIE [46], FLIGHT-Net [35], Retinexformer [4].

**Table 1: Quantitative comparison on the LOL-v1 [49], LOL-v2-real [54], LOL-v2-syn [54], LSRW-Huawei [16], and LSRW-Nikon [16]. Boldface indicates the best results, the second-best results are underlined.**

| Methods | LOL-v1 PSNR SSIM | LOL-v2-real PSNR SSIM | LOL-v2-syn PSNR SSIM | LSRW-Huawei PSNR SSIM | LSRW-Nikon PSNR SSIM |
|---|---|---|---|---|---|
| SID [7] | 14.35 0.436 | 13.24 0.442 | 15.04 0.610 | 13.81 0.447 | 13.26 0.384 |
| KinD [58] | 20.86 0.790 | 14.74 0.641 | 13.29 0.578 | 16.58 0.569 | 11.52 0.383 |
| LPNet [28] | 21.46 0.802 | 17.80 0.792 | 19.51 0.846 | 15.79 0.546 | 14.61 0.375 |
| Band [53] | 20.13 0.830 | 20.29 0.831 | 23.22 0.927 | 16.63 0.574 | 16.14 0.443 |
| IPT [8] | 16.27 0.504 | 19.80 0.813 | 18.30 0.811 | 18.12 0.517 | 15.08 0.380 |
| Sparse [54] | 17.20 0.640 | 20.06 0.816 | 22.05 0.905 | 17.34 0.542 | 14.73 0.396 |
| HDMNet [30] | 23.45 **0.852** | 18.55 0.713 | 20.54 0.854 | 20.81 0.607 | 16.65 0.487 |
| MIRNet [56] | 24.14 0.830 | 20.02 0.820 | 21.94 0.876 | 19.98 0.609 | 17.10 0.502 |
| SNR-Net [52] | 24.61 0.842 | 21.48 0.849 | 24.14 0.928 | 20.67 0.591 | 17.54 0.482 |
| FourLLIE [46] | 24.15 0.840 | 22.34 0.847 | 24.65 0.919 | 21.30 0.622 | 17.82 0.504 |
| FLIGHT-Net [35] | 24.96 0.850 | 21.71 0.834 | 24.92 0.930 | 20.65 0.623 | 16.97 0.471 |
| Retinexformer [4] | 25.16 0.845 | 22.80 0.840 | 25.67 0.930 | 21.37 0.631 | 18.06 0.517 |
| Ours | **25.78 0.852** | **23.56 0.853** | **26.27 0.938** | **22.89 0.644** | **19.12 0.539** |

(a) Low-light    (b) SNR-Net [52]    (c) FourLLIE [46]    (d) FLIGHT-Net [35]    (e) Retinexformer [4]    (f) Ours    (g) Ground Truth

**Figure 8: Qualitative comparison on LOL-v1 [49], LOL-v2-real [54], and LOL-v2-syn [54] (top to bottom). Our method is closer to the illumination distributions of ground truth.**

**Quantitative comparison.** We use peak signal-to-noise ratio (PSNR) and structural similarity (SSIM) [47] as evaluation metrics. In general, the higher PSNR and higher SSIM indicate that the quality of the generated image is better and more similar to ground truth. It is worth noting that we either obtain the values from respective authoritatively published papers or run the respective publicly released codes to obtain these values.

Table 1 shows the comparison of LOL-v1, LOL-v2-real, LOL-v2-syn, LSRW-Huawei and LSRW-Nikon. From the quantitative results, our method significantly outperforms the comparative models for low-light image enhancement in terms of PSNR and SSIM, demonstrating the advantages of our proposed method. In particular, the higher SSIM indicates more high-frequency information and structure in the results. Undoubtedly, our method retains the high-frequency information well. Notably, compared to the recent state-of-the-art method Retinexformer, our method achieves improvements of 0.62, 0.76, 0.6, 1.52, and 1.06 dB on five datasets,

respectively. Significantly, our method is closer to the illumination distribution of ground truth.

**Qualitative comparison.** We give the qualitative comparison of LOL-v1, LOL-v2-real and LOL-v2-syn in Fig. 8. Previous methods on the LOL-v1 dataset either lost color information (like SNR-Net ) or have low illumination in local sub-regions (like FourLLIE and FLIGHT-Net).The visualization results of the LOL-v2-real dataset show that previous methods cannot effectively suppress noise (like SNR -Net and FLIGHT-Net).The visualization results of the LOL-v2-syn dataset show that ignoring the illumination information (like FLIGHT-Net) results in image distortion. However, paying too much attention to the illumination information (like Retinexformer) results in an overly bright image. In contrast, our method shows higher contrast, more accurate illumination distribution and more detailed texture structure to achieve the best visual effect.

Fig. 9 shows the qualitative comparison of LSRW-Huawei and LSRW-Nikon, respectively. On the LSRW-Huawei dataset, previous

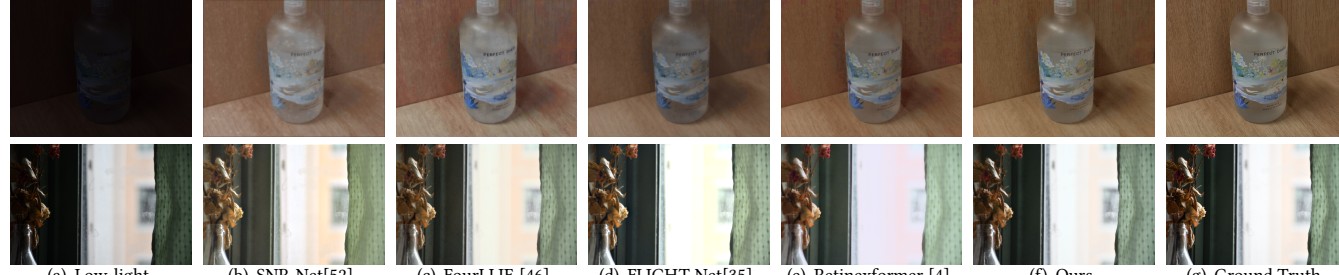

| (a) Low-light | (b) SNR-Net[52] | (c) FourLLIE [46] | (d) FLIGHT-Net[35] | (e) Retinexformer [4] | (f) Ours | (g) Ground Truth |

**Figure 9: Qualitative comparison on LSRW-Huawei [16] (top) and LSRW-Nikon [16] (bottom). Our method recovers detailed textures better.**

**Table 2: Results of the ablation study. The best results are boldfaced and the second-best ones are underlined.**

| Methods | LOL-v1 PSNR SSIM | LOL-v2-real PSNR SSIM | LOL-v2-syn PSNR SSIM | LSRW-Huawei PSNR SSIM | LSRW-Nikon PSNR SSIM |
|---|---|---|---|---|---|
| Ours w/o DFFN | 21.28 0.786 | 19.96 0.784 | 21.89 0.879 | 18.76 0.548 | 16.89 0.461 |
| Ours w/o Transformer | 22.86 0.804 | 21.78 0.823 | 24.74 0.923 | 20.93 0.619 | 17.86 0.503 |
| Ours w/o DWT | 23.17 0.821 | 21.41 0.817 | 24.96 0.925 | 19.98 0.594 | 17.14 0.481 |
| Ours w/o IDP | 23.05 0.818 | 21.62 0.821 | 24.31 0.918 | 20.61 0.607 | 17.35 0.499 |
| Ours | **25.78 0.852** | **23.56 0.853** | **26.27 0.938** | **22.89 0.644** | **19.12 0.539** |

methods either produce blurred images (like SNR-Net), fail to recover detailed textures (like FourLLIE), or lost illumination information (like FLIGHT-Net). On the LSRW-Nikon dataset, FLIGHT-Net and Retinexformer cause color distortion and noise amplification. Compared with other methods, our method is closer to the ground truth distribution and represents clearly for some complex textures.

## 4.3 Ablation Study

In order to verify the rationality of the module setup in our network architecture, we remove different components of the architecture to perform ablation experiments. **1)** "Ours w/o DFFN" removes the Dual-domain Feature Fusion Network. **2)** "Ours w/o Transformer" removes the architecture of transformer in the Dual-domain Feature Fusion Network. **3)** "Ours w/o DWT" removes the role of the discrete wavelet transform in the Dual-domain Feature Fusion Network. **4)** "Ours w/o IDP" removes the Illumination Distribution Prior. Table 2 shows the results of the ablation studies for all five datasets.

"Ours w/o DFFN" demonstrates that we can still obtain images with good illumination conditions only in the presence of an illumination distribution prior. However, for some complex textures, full recovery cannot be obtained. "Ours w/o Transformer" shows that with the illumination distribution prior and the discrete wavelet transform, we not only obtain images with good illumination, but also extract local high-frequency information to maintain structural consistency. Compared with "Ours", "Ours w/o Transformer" ignores the importance of global spatial information. "Ours w/o DWT" shows that the transformer architecture can facilitate the recovery of complex textures by capturing global spatial information on top of the illumination distribution to achieve illumination. However, "Ours w/o DWT" may ignore the effect of local signals. To be

precise, the perturbed local signals interfere with the performance of the transformer to some extent. "Ours w/o IDP" shows that only with the dual-domain feature fusion network, although we can obtain well-performing images, we lack the attention to the illumination information, and the obtained images possess uncertain illumination conditions.

The above analyses demonstrate the limitations of single-domain information in recovering detailed textures, thus suggesting that dual-domain information facilitates feature recovery. Illumination distribution prior guides accurately when recovering image illumination. Overall, our full setup shows the highest PSNR and SSIM among all ablation settings.

## 5 CONCLUSION

In this paper, we have proposed a simple and efficient prior, called illumination distribution prior, for illuminating the darkness of low-light images. The illumination distribution prior is based on the statistics of low-light images and ground truth and the positive correlation of the mean value and standard deviation of the images with the illumination. By combining the illumination distribution prior with the dual-domain information fusion network, we are able to obtain images that are more consistent with the ground truth distribution. The Dual-domain Feature Fusion Network, by virtue of inserting the discrete wavelet transform into the transformer architecture, skillfully extracts both local high-frequency information and global spatial information to facilitate the recovery of image detail texture. Extensive experiments on five low-light image enhancement datasets show that our network architecture has state-of-the-art performance. In the future, we will explore the application of illumination distribution prior in other computer vision fields.

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
