# OpenReview forum: "Illumination Distribution Prior for Low-light Image Enhancement"
_acmmm.org/ACMMM/2024/Conference — MM2024 Poster_

### Official Review · Reviewer_nKp4 · 2024-05-22

**Rating:** 5
**Confidence:** 3

**Summary:**

This paper introduces an illumination distribution prior (IDP) for enhancing low-light images. By incorporating IDP into the dual domain feature fusion network (DFFN), the resulting images closely match the ground truth distribution, thus improving the quality of low-light image enhancement.

**Strengths:**

1. The illumination distribution prior for low-light image enhancement is intriguing to this work.
2. The analysis of the illumination distribution prior in the experiments is convincing to me.

**Limitations:**

1. The visual results of this work in Figure 8 do not appear to be particularly pronounced.
2. The authors can add more objective metrics in Table 1, not only PSNR and SSIM.
3. It lacks comparison of computational complexity.

**Suitability:**

3

---

### Official Review · Reviewer_Wh95 · 2024-05-25

**Rating:** 3
**Confidence:** 3

**Summary:**

This manuscript proposes a simple but effective illumination distribution prior (IDP) for images to illuminate the darkness. The illumination distribution prior is the product of a statistical approach to low-light images. Using IDP combined with the dual domain feature fusion network (DFFN), the work can obtain images that are more consistent with the ground truth distribution. DFFN inserts the discrete wavelet transform (DWT) into the transformer architecture, aiming to recover the detailed texture of the image through local high-frequency information and global spatial information.

**Strengths:**

The theoretical analysis in the manuscript is sound.

**Limitations:**

(1) Ablation study is insufficient. Merely removing some parts does not demonstrate the effectiveness of the proposed structure. Additionally, as the focus of the manuscript, the design of preprocessing seems to lack necessary combined experiments and independent validation.
(2) According to the experiments in the manuscript, the model's performance experienced a significant decrease in the "w/o IDP" condition. Does this indicate that when the same preprocessing is applied, other comparative methods may outperform the method proposed in this manuscript?
(3) In digital image processing, many classical methods are similar to IDP. The article seems to lack a comparison with such methods.

**Suitability:**

2

---

### Official Review · Reviewer_whVt · 2024-05-25

**Rating:** 3
**Confidence:** 3

**Summary:**

This paper proposes a novel illumination distribution prior for lowlight image enhancement by constructing the multi-scale illumination distribution migration. In this work, they design a dual-domain feature fusion network, which integrates the local frequency-domain information of the discrete wavelet transform and the global spatial information of the transformer architecture, and utilizes the rich dual-domain information for fine-grained restoration of texture structure and noise removal.

**Strengths:**

The motivation behind the proposed method is clear.

**Limitations:**

(1) Some spelling errors need to be corrected.
(2) The novelty of the proposed method seems to be limited, as both the network architecture and the approach of combining dual domains appear to be quite common in the field of image restoration.
(3) The experiments are not extensive enough. For instance, there is a lack of experiments regarding the combination of IDP with other comparative methods.

**Suitability:**

2

---

### Meta-Review · Program_Chairs · 2024-07-06

**Recommendation:** Accept (Poster)
**Confidence:** 4

**Metareview:**

The authors propose a illumination distribution prior (IDP) for images to illuminate the darkness. The distribution prior is the product of a statistical approach to low-light images. The method uses a a dual-domain feature fusion network, integrating the local frequency-domain information of the discrete wavelet transform and the global spatial information of the transformer architecture. The method is technically sound and an interesting approach. While the novelty and the experimental validation is somewhat limited, with more tests and analysis required for the validation of the method, given the overall positive ratings from reviewers, this paper is recommended for accept.